# The Assessment of Natural Vaginal Delivery in Relation to Pregnancy-Related Anxiety—A Single-Center Pilot Study

**DOI:** 10.3390/healthcare11101435

**Published:** 2023-05-15

**Authors:** Anna Michalik, Michalina Pracowity, Lucyna Wójcicka

**Affiliations:** Department of Obstetric and Gynecological Nursing, Faculty of Health Sciences with the Institute of Maritime and Tropical Medicine, Medical University of Gdansk, Dębinki 7, 80-211 Gdańsk, Polandlucyna.wojcicka@gumed.edu.pl (L.W.)

**Keywords:** vaginal birth, fear of childbirth, pregnancy-related anxiety

## Abstract

Background: Pregnancy-related anxiety (PrA) is a specific type of anxiety experienced during the perinatal period. It may concern a person’s health and physical appearance, fetal development, hospital and health care experiences, impending childbirth, and early parenthood. PrA is considered to be a stronger predictor of adverse pregnancy outcomes than general anxiety and depression. The purpose of this research was to conduct a pilot study and evaluate the course of vaginal birth (VB) in relation to PrA levels in a population of pregnant women with low obstetrical risk. Methods: This cross-sectional exploratory study included 84 pregnant women (with a mean age of 28.61 ± 4.99) (without cesarean section (CS) indications and with a low risk of complications during VB). Research questionnaires were distributed and filled in in person during the course of hospitalization. Groups that varied at the level of PrA were compared using the Wilcoxon rank-sum test, Fisher’s exact test, or chi-square test of independence, as appropriate. Results: More than two-thirds of the respondents (72.6%) were medicated in labor. Women with high PrA, selected based on a cut-off point with a total PRAQ-R2 score of 60, experienced significantly longer first (start of established labor to fully dilated cervix) and second (lasts from when cervix is fully dilated until the birth) periods of labor, instrumental delivery, or emergency CS. In the group with high PrA levels, a episiotMmentation of evidence-based recommendations for the affected population to identify and further treat women with elevated levels of PrA.

## 1. Introduction

Anxiety is defined as an adverse emotional state associated with the anticipation of danger from outside or within the body, manifesting as uneasiness, tension, restraint, and threats [1]. During pregnancy, the intensity and type of emotional involvement of women are unique. Understandably, pregnancy may involve vulnerability to developing psychological distress. Pregnancy-related anxiety (PrA) or pregnancy-specific anxiety (PSA) are terms that define a particular type of anxiety experienced during the perinatal period. It may concern a person’s health and physical appearance, fetal development, hospital and health care experiences, impending childbirth, and early parenthood [2,3]. PrA, its characteristics, its impact on the course of pregnancy, childbirth and early parenthood, and predisposing/triggering factors have become important areas of research over the past 15 years. We have also gained knowledge on how to identify and measure PrA. The distinct nature of PrA is emphasized because it is modestly associated with standard scales for measuring anxiety in general and can be separately characterized in relation to external correlates [4,5,6,7,8].

Fear of childbirth (FOC) is a specific type of PrA [6]. It has been reported that 9–35% of women experience severe FOC, which correlates with many possible complications [7,9]. It is well documented that PrA and FOC correlate with prolonged durations of active labor, the greater use of pain relief, higher rates of emergency cesarean sections (CSs), higher rates of cesarean sections without medical indications, the induction of labor, and other obstetric interventions [9,10,11,12,13,14,15,16,17]. It has also been proven that PrA is associated with negative personal birth experiences, adversely affects mother–child bonding, impedes effective breastfeeding, and may be a predisposing factor in the development of postpartum depression and even postpartum psychosis [3,5,6,18]. A mother who has developed PrA, causing potential harm to her infant during childbirth, is quite different from one who has mood disorders and depressive symptoms, causing treatment difficulties. These emotional states and their predisposing and sustaining factors can be distinguished and may co-occur [3]. The most relevant mechanism that seems to explain the relationship between maternal prenatal psychosocial stress, anxiety, and pregnancy and childbirth complications is the activity of the hypothalamic–pituitary–adrenal axis (HPA axis), with cortisol as its end-product. The fetus must be exposed to cortisol due to various aspects of brain development and late gestational lung maturation. Nevertheless, it is believed that exposure to high cortisol levels in utero can adversely affect pregnancy and delivery, as well as the behavioral, immunological, and brain development of the fetus [9].

PrA is considered to be a stronger predictor of adverse pregnancy outcomes than general anxiety and depression [3,5]. Reports indicate that high PrA and FOC may correlate with the risk of preterm delivery and low neonatal birth weight, and negatively affect neonatal neurological and behavioral development [5,19,20,21,22]. It is noteworthy that most available studies indicate that adverse effects on offspring via fetal programming implicate anxiety (not depression) [4,21]. It has also been demonstrated that PrA, depression, and stress are directly linked to microbiota disturbances in pregnant women and newborns [11]. Additionally, it has been found that there is an association between PrA and an engagement with behaviors that affect health (e.g., excessive weight gain during pregnancy, smoking, etc.) [2,8].

There are noticeable differences in PrA rates between primiparous and multiparous women [17,21]. PrA and FOC are more likely to correlate with perinatal experiences in multiparous women. Sudden cesarean sections and instrumental deliveries are particularly traumatic [8,21,22].

The pregnancy-related anxiety diagnosis process and criteria are not standardized. Although the phenomenon is universal and occurs among women all over the world, the factors correlating with PrA vary depending on individual, social, cultural, and ethnic factors. Polish perinatal care involves an exceptionally high rate of cesarean sections (47% in 2022) [13], and the reason for this phenomenon remains unknown. In addition, epidural availability in Poland is currently insufficient, and only 35–40% of women in labor have access to this procedure [14]. Interestingly, the highest CS rates are reported in the regions of Poland where epidural access is the most limited due to an insufficient number of anesthesiologists. The factors mentioned are likely to contribute to elevated PrA and FOC rates.

Recently, increasingly more research has started to explore the correlation between elevated levels of labor anxiety and pregnant women’s perinatal preferences, as well as the impact of those levels on the course of natural vaginal delivery in Polish perinatal practice. Preliminary reports indicate that experiencing elevated levels of FOC unequivocally correlates with the cesarean section being selected as the preferred delivery method in healthy pregnant women (with low risk of perinatal complications and no existing indications for cesarean sections) [15]. The purpose of this research was to conduct a pilot study and evaluate the course of vaginal delivery in relation to PrA levels in a population of pregnant women with low obstetrical risk for vaginal birth complications.

## 2. Materials and Methods

The study was conducted in the first quarter of 2022. To determine the minimum sample size in the initial study, a recommended on-line statistical tool was used (raosoft.com) [16]. The initial assumptions for calculations were based on available statistical data: the average number of natural births in a month in 2021 in the tested facility (181 births) and reported FOC experiences in a female population (9–36% and 7%) [7]. The accepted margin of error was 5%, the confidence level was 95%, and the response distribution was 9–36%. The recommended sample size was estimated at 75–120 respondents. The final study group included 84 pregnant women (without cesarean section indications and with a low risk of complications during natural vaginal delivery) with pregnancies over 35 weeks, reporting to the University Hospital for delivery. The sample size was considered as fair for the pilot study.

Research questionnaires were distributed and filled in direct contact in the course of hospitalization prior to labor. The exclusion criteria were as follows: an active stage of labor (contractions are regular and cervix has dilated to 6 cm), gestational age below 35 weeks, indications for cesarean, a high level of potential risk for vaginal birth complications (estimative birth weight more than 4200 g, breech presentation, and preeclampsia), personal preference for cesarean and diagnosed depression, and anxiety or mental health issues. According to Polish recommendations, a prior cesarean section is not a contraindication to vaginal birth, so 9 women after CS were included in the study group. All respondents were informed about the purpose of the study and the planned method of publishing the results, and consented to participate in this research.

The Independent Bioethics Committee for Scientific Research at the Medical University of Gdansk approved the protocol of this study.

### 2.1. Research Tools

A cross-sectional proprietary questionnaire was used to obtain the participants’ background information. The form consisted of 25 questions and allowed for demographic data and obstetric information to be collected, such as comorbidities during pregnancy (diagnosed by a physician), gestational week, parity, obstetric history, and participation in antenatal classes.

The study utilized the Pregnancy-Related Anxiety Questionnaire—Revised 2 (PRAQ-R2), proposed by Huizink [17], the Polish version by Michalik et al. [18]. The tool has been universalized for all pregnant women, regardless of parity. It is a self-reporting measure for patients, but it can also be used during an interview with a healthcare professional. The PRAQ-R2 consists of 10 questions grouped into 3 subscales: fear of giving birth (FoGB; items 1, 2, and 6); worries of bearing a physically or mentally disabled child (WaHC; items 4, 8, 9, and 10); and concerns about own appearance (CoA; items 3, 5, and 7) [17,18]. The total score ranges from 10 to 50 points. Higher scores suggest greater PrA. No clinical cut-off point was defined for this questionnaire.

Another questionnaire was used for the retrospective evaluation of labor, the duration of its phases, as well as labor pain relief methods and complications.

### 2.2. Statistical Analyses

The results for the continuous variables are expressed as the mean standard deviation (SD), while the results for the dichotomous variables are expressed as frequencies (%). Correlations between PRAQ-R2 scores and other variables such as labor duration or newborn wellness were evaluated using Spearman’s rho coefficient. The differences in the total mean PRAQ-R2 score between nulliparous and parous groups were evaluated with the (non-parametric) Mann–Whitney U test. The same test was used to compare the analyzed variable in other observation groups, and was separated based on characteristics such as marital status, education, or the use of labor analgesia. Comparative analysis was performed using the Kruskal–Wallis test to assess the association of PrA with the final mode of delivery. Groups of women varied at the level of PrA were compared using the Wilcoxon rank-sum test, Fisher’s exact test, or the chi-square test of independence, as appropriate. The Shapiro–Wilk tests were used to verify how close the data collected with the PRAQ-R2 form fit to a normal distribution. To confirm the internal consistency of the information gathered by the PRAQ-R2, Cronbach’s alpha values were calculated. After Bonferroni correction, the original acceptable α level was adjusted from α < 0.05 to α < 0.005.

Statistical analyses were performed using the R software and statistics calculators from Statistics Kingdom (http://www.statskingdom.com/index.html (accessed on 23 October 2022)).

## 3. Results

### 3.1. Descriptive Statistics

The mean age of participants was 28.61 ± 4.99 years. Most respondents lived in cities (85.7%), had a college or university degree (59.5%) and secondary (23.8%) education, and were married or cohabitating (59.5%). The mean gestational age was 38.2 ± 2.79 weeks, while the mean number of pregnancies at the time of the study was 2.06 (including the present one). Women in their first pregnancy (41.7%) and primiparas (53.6%) constituted the most numerous subgroup in the study.

Complications in the current pregnancy occurred in 32.1% of respondents. The most common pregnancy complications included gestational diabetes (41.4%) and gestational hypertension (31%), followed by the risk of premature delivery (27.6%), hypothyroidism (27.6%), and bleeding (24.1%).

For all respondents, the preferred method of delivery was vaginal birth; for 85.7% of them, it was the final mode of delivery. Detailed demographic and obstetric characteristics are presented in Table 1.

The majority of respondents did not experience labor complications (79.8%). The most common complications included a lack of labor progress, hemorrhages, or instrumental delivery (vacuum). More than two-thirds of the respondents (72.6%) were medicated in labor. Labor analgesia was used in 56% of the surveyed women, and the perineal incision was performed in as many as 54.2% of respondents.

The average level of anxiety in the study group was 30.75 ± 8.56 points, as assessed by the total PRAQ-R2 score. The scores ranged between 10 and 50 points, with more than half of the respondents (56%) scoring higher or equal to 30 points, which is 60% of the maximum total score. Based on the characteristics of the Visual Analogue Scale (VAS) [19], which is another popular measurement scale applied for validating the parametric properties of the PRAQ-R2; this value was taken as an intuitive cut-off point and a score of ≥30 points was defined as a high level of PrA. To confirm the validity of the adopted data cut-off point, Shapiro–Wilk tests were conducted and they did not show significant abnormalities (W(84) = 0.989) (*p* = 0.718). Consequently, the majority of cases were close to the average score, and the adopted cut-off range will increase the chance of identifying women who require further intervention among the subjects.

The most frequently indicated source of anxiety was the pain experienced during childbirth. The fear of labor itself was the second, followed by the fear that the child would be born with a defect or suffer from physical damage, or that the child would be mentally disabled or suffer brain damage. A small percentage of women experienced anxiety related to significant weight gain and an unattractive appearance. Descriptive statistics for the PRAQ-R2 questionnaire are presented in Table 2.

The study also included an analysis of how PrA affected the duration of delivery. In the study group, the first period (the start of established labor to fully dilated cervix) of labor was 393.17 min (6.55 h) on average, the second period (from when cervix is fully dilated to the birth) was 58.43 min, and the third period (from when the baby is born to when the placenta comes out) was 13.06 min on average. Spearman’s rho correlation showed a strong relationship between the duration of each stage of labor and PrA (ρ 0.69–0.63). The relationships were positive, which means that women with increased PrA experienced significantly longer first and second periods of labor. In the group of women with high PrA, which was selected based on a total PRAQ-R2 cut-off point score of 60%, the average duration of the first period of labor was 7 h and 42 min, which was 156 min longer than in the group of women with low PrA. Similar correlations were observed in the second and third periods of labor, which were longer by 25 min and 5 min, respectively, in the group with high PrA levels (Table 3).

Statistical tests were performed to assess statistically more variables associated with PrA. The level of anxiety among multiparous and primiparous participants was compared using the Mann–Whitney U test. The analysis results were statistically significant (Z = 5.43; *p* < 0.001). Women who gave birth for the first time had a significantly higher anxiety score in relation to childbirth. The marital status of the respondents was also associated with the severity of fear of childbirth (Z = 2.08; *p* < 0.05). Unmarried women experienced higher levels of fear of childbirth. Moreover, women who received labor analgesia had a statistically significant (Z = 4.25; *p* < 0.001) higher intensity of labor anxiety. However, Mann–Whitney U-test analyses showed no association of the level of anxiety before childbirth with education (Z = 0.17; *p* = 0.862) or attending childbirth school (Z = 0.26; *p* = 0.798).

Another studied aspect was whether the level of labor anxiety correlates with the frequency of surgical termination of labor and emergency cesarean. For this purpose, comparative analysis was performed using the Kruskal–Wallis test, and the result was found to be statistically significant (χ2 = 25.03; *p* < 0.001). The respondents whose labor ended with an instrumental delivery or cesarean section had higher PrA than women delivering through vaginal birth. A similar situation was observed when a perineal incision was required during the second period of labor. The Mann–Whitney U test shows a significant association between the PrA level and the need for a perineal incision (Z = 10.6; *p* < 0.001). In the group with high PrA levels, a perineal incision was performed almost twice as often.

We also investigated whether the occurrence of labor anxiety affects the condition of the newborn after birth. For this purpose, Spearman’s rho correlation analysis was performed, which showed that this relationship was statistically significant (ρ = −0.61; *p* < 0.001). The relationship between labor anxiety and the infant’s Apgar score was found to be negative, which means that in the group of respondents with higher levels of labor anxiety, the newborns scored lower on the Apgar Scale. Pearson correlation results indicate that there was also a significant minimal negative relationship between age and the total PRAQ-R2 score (r (82) = 0.345, *p* = 0.001).

Bonferroni correction was used to avoid type I errors and potentially rejecting the null hypothesis that is actually true since multiple tests were conducted simultaneously to compare several groups. The original acceptable α level was adjusted from α < 0.05 to α < 0.005. The main analyzed variables statistically associated with PrA are presented in Table 4.

In the final stage of the study, the participants were divided into two groups based on a total PRAQ-R2 score ≥30 points with a cut-off point of 60%: women with a low PrA level and women with a high PrA level. (Table 5) The groups were compared based on the previously analyzed associations of PrA with other variables. The tests performed confirmed statistically significant (*p* < 0.05) differences between women with high and low PrA levels. The first differentiating variable was the age of subjects. The observed common language effect size, U1/(n1n2), is 0.68, i.e., it is probable that a random age value from the low-PrA-level group is greater than a random value from the high-PrA-level group. The observed standardized effect size, Z/√(n1 + n2), is medium (0.31). This indicates that the magnitude of the difference between the value from the low-PrA-level group and the value from the high-PrA-level group is medium. Detailed data on the comparison of women’s groups and other significant differences found between them are presented in Table 6.

### 3.2. Reliability

Cronbach’s alpha coefficient was the basis for assessing the internal consistency of the PRAQ-R2 in the current study. Cronbach’s alpha was good (0.86) for the total construct and good or acceptable for the three subscales. Cronbach’s alpha was 0.76 for the FoGB subscale, 0.89 for WaHC, and 0.82 for CoA, which is in line with the results from the Polish version of the PRAQ-R2 proposed by Michalik et al. [18], where Cronbach’s alpha internal consistency coefficient in subscales ranged from 0.68 to 0.91. Corrected item–total correlations were moderate-to-high in subscales and in the total construct; all values ranged from 0.48 to 0.84, i.e., above the recommended item selection value (≥0.20). Cronbach’s Alpha if Item Deleted values were less than 0.86. All items were retained. Table 6 presents the correlation between the individual items and the overall score.

## 4. Discussion

In our study, elevated PrA and its consequences were found in primiparas, single (cohabitating but not married) women, and urban dwellers. The average age in this group was 27 (Polish women deliver their first child at 27 on average).

In the described group, a relationship was observed between elevated PrA levels and a longer duration of the first and second periods of labor, a higher frequency of epidural use, and a higher frequency of labor termination through emergency cesarean and instrumental birth (vacuum and forceps). In the high-PrA-level group, perineal incision was performed almost twice as often. In addition, newborns had lower Apgar scores immediately after delivery in the high-PrA-level group.

The pregnancy-related anxiety diagnosis process and criteria are not standardized. Although the phenomenon is universal and occurs in women around the world, the factors correlating with PrA vary according to individual and social characteristics, and may be culture- or ethnicity-specific [3,20,21,22,23]. For example, reports from Canada and the United States show that women are at the highest risk of experiencing PrA if they are delivering their first child, are unmarried, have been treated for infertility, have experienced a previous miscarriage, are in a high-risk group due to medical conditions, have low income, are of Hispanic origin, or have had a previous anxiety disorder [20]. Research from Finland, in turn, shows that higher levels of anxiety were reported among women who had an advanced maternal age, were single, had a high and unspecified socioeconomic status, and suffered from depression, regardless of whether they were multiparous or primiparous [21]. Most studies show that younger women experience more PrA, but women of advanced maternal age are sometimes more anxious than younger women, suggesting a curvilinear pattern with maternal age [3]. The results from our previous study on childbirth preferences are consistent with reports on the associations between previous VB experiences and PrA in a subsequent pregnancy. In the study, in multiparous women, PrA was most common after one previous VB, reflecting the importance of experience as a predisposing factor [15]. Other reports have also shown the relationship between elevated PrA levels and primiparity and previous negative obstetric experiences, previous emergency cesarean, and instrumental birth (vacuum and forceps) [3,12,14]. Our results are consistent with studies that highlight the prolongation of the first and second phases of labor in groups of women with significantly elevated PrA [22,23,24].

## 5. Conclusions

Since it is well documented that PrA and FOC correlate with significant complications of pregnancy, labor, and the postpartum period, as well as the early childhood period, good obstetric practice calls for the need to implement evidence-based recommendations for the affected population in order to identify and further treat women with elevated levels of anxiety [9,19]. The World Health Organization (WHO) recommends introducing additional psychoeducation for women with elevated PrA and FOC as a non-clinical recommendation to reduce unnecessary cesarean sections [23]. This is a valid solution given that PrA is a stronger predictor of adverse pregnancy outcomes than general anxiety and depression.

Additionally, we found that the current study results on the internal consistency of the PRAQ-R2 are similar to the results shown in the Polish version of the PRAQ-R2 proposed by Michalik et al. [18]. The whole construct and all three subscales showed high internal consistency, with Cronbach’s alfa coefficient values ranging from 0.48 to 0.84. Analyses showed that the reliability of the Polish version of the PRAQ-R2 is high and the tool functions properly. We recommend the PRAQ-R2 questionnaire for PrA assessment among polish women.

Future research is needed to demonstrate the connection between PrA levels among low-risk pregnant women and the number of caesareans on demand in Polish perinatal practice. We hypothesize that the CS rate at a level of 47% among Polish women can be linked with high PrA and FOC factors. These data are necessary in order to describe and assess Polish perinatal practice in terms of current recommendations and evidence-based knowledge.

### Limitations of the Study

A cut-off level of 60% for the Polish adaptation of the PRAQ-R2 R2, as proposed by Michalik et al. [18], has theoretical implications. There is an additional need to continue this research on wider populations of Polish women.

Furthermore, there is a significant need to continue this research on a wider multicentered population.

## Figures and Tables

**Table 1 healthcare-11-01435-t001:** Demographic and obstetric characteristics of the study population.

Variable	Mean ± SD	N	n%	Variable	Mean ± SD	N	n%
Age (years)	28.61 ± 4.99	84	100	Place of residence			
19–25		28	33.3	Rural		12	14.3
26–30		21	25.0	City < 100 thous. inhab.		38	45.2
31–35		28	33.3	City 10–100 thous. inhab.		27	32.2
36–40		7	8.4	City over 10 thous. inhab.		7	8.3
Civil status				Educational level			
Single		33	39.3	Vocational		14	16.7
Married or cohabiting		50	59.5	High school		20	23.8
Divorced		1	1.2	University		50	59.5
Number of pregnancies	2.06 ± 1.25			Gestational age (weeks)	38.2 ± 2.79		
First		35	41.7	≥35		9	10.7
Second		26	31.0	36–37		9	10.7
Third		13	15.5	38–39		38	45.3
Consecutive		10	11.8	40–41		28	33.3
Number of childbirths	1.65 ± 0.91			Participation in antenatal classes			
First		45	53.6	Yes		40	47.6
Second		29	34.5	Unfortunately not		25	29.8
Consecutive		10	11.9	No		19	22.6
Planned pregnancy				Mode of delivery in a previous pregnancy *			
Yes		68	81.0	Vaginal birth		30	76.9
No		16	19.0	Cesarean section		9	23.1
Gestational week of previous pregnancy termination *	38.3 ± 2.42			Subjective assessment of previous labor *			
≥35		3	7.7	Very good		9	23.1
36–37		7	17.9	Quite good		14	35.9
38–39		17	43.6	Not so well		11	28.2
40–41		12	30.8	Very bad		5	12.8
Complications during previous labor *				Complications during pregnancy			
Yes		27	31.6	Yes		27	32.1
No		60	71.1	No		57	67.9
Multitude of pregnancy				Childbirth complications			
Single		82	97.6	Yes		17	20.2
Twin/multiple		2	2.4	No		67	79.8
Planned mode of delivery				Final mode of delivery			
Vaginal birth		84	100	Vaginal birth		72	85.7
Cesarean section		0	0	Cesarean section		12	14.3

* (multiparous only); SD (standard deviation).

**Table 2 healthcare-11-01435-t002:** Assessment of pregnancy-related anxiety with the PRAQ-R2.

PRAQ-R2 Items	Min	Max	M	SD	Me
1. I am anxious about the delivery.	1	5	3.58	1.24	4
2. I am worried about the pain of contractions and the pain during delivery.	1	5	3.60	1.32	4
6. I am worried about not being able to control myself during labour and fear that I will scream.	1	5	2.77	1.34	3
Subscale: Fear of giving birth (FoGB)	3	15	9.95	3.19	10
4. I sometimes think that our child will be in poor health or will be prone to illnesses.	1	5	3.13	1.40	3
8. I am afraid the baby will be mentally handicapped or will suffer from brain damage.	1	5	3.32	1.23	4
9. I am afraid our baby will be stillborn, or will die during or immediately after delivery.	1	5	3.25	1.22	3
10. I am afraid that our baby will suffer from a physical defect or worry that something will be physically wrong with the baby.	1	5	3.33	1.24	4
Subscale: Worries about bearing a handicapped child (WaHC)	4	20	16.36	5.54	16.5
3. I am worried about the fact that I shall not regain my figure after delivery.	1	5	2.70	1.33	3
5. I am concerned about my unattractive appearance.	1	5	2.56	1.26	3
7. I am worried about my enormous weight gain.	1	5	2.50	1.32	2
Subscale: Concern about own appearance (CoA)	3	15	7.76	3.34	8
Total score	10	50	30.75	8.56	31

Min—minimum, Max—maximum, M—mean, SD—standard deviation, Me—median.

**Table 3 healthcare-11-01435-t003:** Spearman’s rho correlation between the duration of labor (minutes) and PrA.

Stage of the Delivery	Population	Min	Max	M	SD	Me	Pregnancy Related Anxiety
I	Total population	130	690	393.17	141.76	375	0.69 ***
	Hihg PrA	140	690	462.02	130.65	465	
	Low PrA	130	675	305.76	113.36	293.5	
II	Total population	7	155	58.43	36.64	48	0.67 ***
	Hihg PrA	26	155	65.45	45.18	60	
	Low PrA	7	131	40.05	29.66	32	
III	Total population	2	48	13.06	10.24	10	0.63 ***
	Hihg PrA	5	48	13.43	12.19	60	
	Low PrA	2	37	8.35	7.14	5	

*** *p* < 0.001, Min—minimum, Max—maximum, M—mean, SD—standard deviation, Me—median.

**Table 4 healthcare-11-01435-t004:** Correlations between PrA and selected variables.

Variable		N (%)	PrA		Comparison Result *	*p*
			*M*	*SD*		
Anesthesia during childbirth	No	37 (44)	28.32	7.18	4.25 ^1^	0.000 ***
	Yes	47 (56)	36.64	9.07		
Apgar Scale	1–3	1 (1.2)	37.00	1.77	−0.61 ^2^	<0.001 ***
	4–7	11 (13.1)	38.82	4.81		
	8–10	72 (85.7)	29.43	8.42		
Mode of delivery	Vaginal birth	67 (79.7)	30.16	8.07	25.03 ^3^	0.000 ***
	Cesarean section	12 (14.3)	43.50	6.50		
	Instrumental vaginal birth (vacuum and forceps)	5 (6.0)	41.00	6.77		
Parity	Primiparous	45 (53.6)	37.91	7.51	5.43 ^1^	0.000 ***
	Multiparous	39 (46.4)	27.28	7.66		
Civil status	Single/divorced	34 (40.5)	35.44	7.98	2.08 ^1^	0.037 **
	Married or cohabiting	50 (59.5)	31.30	9.72		
Educational level	Vocational/high school	34 (40.5)	33.26	10.36	0.17 ^1^	0.862
	University	50 (59.5)	32.78	8.48		
Perineal incision	Yes	40 (54.8)	30.09	8.72	10.6 ^1^	0.000 ***
	No	33 (45.2)	1.45	0.5		
Participation in antenatal classes	Yes	40 (47.6)	33.00	8.50	0.26 ^1^	0.798
	No	44 (52.4)	32.95	9.95		

M—mean, SD—standard deviation, *p*—statistical significance level, * performed tests: ^1^ Mann–Whitney U test, ^2^ Spearman’s rho correlation, ^3^ Kruskal–Wallis test, ** *p* < 0.05, *** *p* < 0.005.

**Table 5 healthcare-11-01435-t005:** Comparison of groups of women with a high and low levels of PrA (study population N = 84).

	Assumed Low Level of PrA		Assumed High Level of PrA						*p* Value ^2^
Variable	Total Score <30 Points n% = 44		60% Total Score ≥30 Points n% = 56		80% Total Score ≥40 Points n% = 16.67		90% Total Score ≥45 Points n% = 4.76		
Age (years)		37 (100) ^1^		47 (100) ^1^		14 (100) ^1^		4 (100) ^1^	0.004498 **
Min/max	23/39		19/39		19/34		19/23		
M	30.43		27.17		25.79		21.5		
SD	4.28		5.13		4.87		1.73		
Me	31		26		25		22		
Place of residence									0.65
Rural		6 (16.22)		6 (12.77)		3 (21.43)		0 (0)	
City		31 (83.78)		41 (87.23)		11 (78.57)		4 (100)	
Civil status									0.007453 **
Single/divorced		9 (24.32)		25 (53.19)		6 (42.86)		3 (75)	
Married/cohabiting		28 (75.68)		22 (46.81)		8 (57.14)		1 (25)	
Educational level									0.711
Vocational		5 (13.51)		9 (19.15)		6 (42.86)		2 (50)	
High school		10 (27.03)		10 (21.28)		2 (14.28)		1 (25)	
University		22 (59.46)		28 (59.57)		6 (42.86)		1 (25)	
Parity									0.0000019 **
Primiparous		9 (24.32)		36 (76.6)		12 (85.71)		4 (100)	
Multiparous		28 (75.68)		11 (23.4)		2 (14.29)		0 (0)	
Participation in antenatal classes									0.4762
Yes		16 (43.24)		24 (51.06)		5 (35.71)		0 (0)	
No		21 (56.76)		23 (48.94)		9 (64.29)		4 (100)	
Mode of delivery									0.007108 **
Vaginal birth		36 (97.3)		36 (76.6)		9 (64.29)		1 (25)	
Cesarean section		1 (2.7)		11 (23.4)		5 (35.71)		3 (75)	
Anesthesia during childbirth									0.0000174 **
No		26 (70.27)		11 (23.4)		1 (7.14)		0 (0)	
Yes		11 (29.73)		36 (76.6)		13 (92.86)		4 (100)	

Min—minimum, Max—maximum, n_s_—sample size, M—mean, SD—standard deviation, Me—median, *p*—significance level. ^1^ Statistics presented: n_s_ (n_s_%). ^2^ Statistical tests performed: Wilcoxon rank-sum test, chi-square test of independence, Fisher’s exact test; ** *p* < 0.05

**Table 6 healthcare-11-01435-t006:** The internal consistency of the PRAQ-R2.

PRAQ-R2 Item	Fear of Giving Birth (FoGB, Items 1, 2, and 6)	Worries about Bearing a Handicapped Child (WaHC, Items 4, 8, 9, and 10)	Concerns about Own Appearance (CoA, Items 3, 5, and 7)	Corrected Item–Total Correlation	Cronbach’s Alpha if Item Deleted
1	0.57			0.59	0.85
2	0.64			0.56	0.85
6	0.57			0.63	0.84
4		0.6		0.57	0.85
8		0.82		0.61	0.84
9		0.81		0.61	0.84
10		0.84		0.66	0.84
3			0.68	0.48	0.85
5			0.67	0.53	0.85
7			0.66	0.46	0.86
Cronbach’s alpha	0.76	0.89	0.82	Total 0.86	

## Data Availability

Data are available on request due to privacy/ethical restrictions.

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
