# Peer review of "The Assessment of Natural Vaginal Delivery in Relation to Pregnancy-Related Anxiety—A Single-Center Pilot Study"

_healthcare, 2023, doi:10.3390/healthcare11101435_

Round 1
Reviewer 1 Report
This is an interesting and clinically relevant paper that I’ve read with pleasure. There are however some issues in need of further clarification:
1.Abstract, methods, and results: the authors mention 1st, 2nd and 3rd period of labor, however it may not be vclaer for all readers what timer periods are referred to. Please clarify/indicate.
2.Materials and methods: The authors state ”to de termine a minimum sample size”. With which reaerach question / operationalisation in mind was this sample size dertemined? Please clarify/indicate.
3.Methods: I think the figure is better left our, for 2 reasons: 1. The written introduction is already clear, so the figure does not really add, and 2. The study is only about some aspects of the model, namely the relationship between demographics, PRA and a selectionof birth/delivery outcomes. The model may make the reader the reader wrongly assume that this study is about all aspects of the model.
4.Methods, research tools: “a cross-esecrionial, descriptive questionnaire”. This is not correctly formulated. It is a cross-sectional study (the questionnaire in itself is not cross-sectional). And the questionnaire itself is not descriptive, but it’s outcomes are analysed with descriptive statistics. An adjective pronun like “sef developed” would beter fit with “questionnaire”.
5.Results: 59.5% of the saple had Universitty education, which seems quite high. Does this resemble national data on education in Poland?
6.Results / statistics: a binary distinction is made between low lever PRA and high level PRA. Would it not be better to work with continuous outcomes? Please explain why / why not.
7.Results/ statistics: there are many tests done. Should there be some sort of correction for this., like Bonferrroni? Please explain why / why not.
11
Author Response
We sincerely thank you for the review and detailed remarks. We greatly appreciate their relevance. The feedback we received has allowed us to introduce specific changes to the article. We hope that this has improved the quality of the article and will allow for it to be approved.

Reviewer 2 Report
Thank you for providing me with the opportunity to review this paper. Below are some recommendations that need to be addressed in order to strengthen this manuscript.
Abstract: Need to report the mean age (and SD) for the sample.
Introduction:
Page 2 states: "It has also been proven that PrA, depression and stress are directly linked to microbiota disturbances in pregnant women and newborns." - can this be rewritten to state "It has also been demonstrated that PrA, depression and stress are directly linked to microbiota disturbances in pregnant women and newborns" .
Page 2: Remove second 'in' in the following statement : "The purpose of this research was to conduct a pilot study and evaluate the course of vaginal delivery in relation to PrA levels in in a population of pregnant women with low obstetrical risk for vaginal birth complications."
Page 3: The authors state: "Based on available statistical data, the recommended sample size was estimated at 75-120 respondents." However, what was the data analysis and effect size selected to calculate this sample size. This needs to be addressed.
Figure 1. It is not clear what the purpose was in providing this Figure given that it is not really discussed. This Figure can probably be removed from the manuscript.
Methods & Materials:
It is not clear whether the researchers obtained information about the participants' previous mental health diagnosis (e.g., previous diagnosis of anxiety), and whether this was examined in the data analyses. If not then this needs to be acknowledged as a limitation in the study.
Results:
PRAQ-R2: Need to report the internal consistency reliability of this measure,. especially in the current study.
Table 3: The correlation values presented in this table does not make sense. Is 0,69 meant to be 0.69?
I am not sure whether Figure 2 is really required. I don't believe it adds very much to the Results section.
Discussion:
Need to note limitations in the study and also suggestions for future research.
Author Response
We sincerely thank you for the review and detailed remarks. We greatly appreciate their relevance. The feedback we received has allowed us to introduce specific, significant changes to the article. We hope that this has improved the quality of the article and will allow for it to be approved.

Reviewer 3 Report
I would suggest putting "Pregnancy-related Anxiety" in the title of the article instead of "fear of childbirth", since the article deals more with anxiety.
Author Response
Thank you for your suggestion – we reformulated the title.